# Utilization of Agroresidues for the Production of Xylanase by *Bacillus safensis* XPS7 and Optimization of Production Parameters

**Shikha Devi** **, Divya Dwivedi and Arvind Kumar Bhatt ***

Department of Biotechnology, Himachal Pradesh University, Summerhill, Shimla 171005, India; shikhasharma003@gmail.com (S.D.); divyadwivedi2502@gmail.com (D.D.)
* Correspondence: bhtarvind@yahoo.com

**Abstract:** The main objective of this study was to investigate the utilization of various agro-industrial wastes such as wheat bran, wheat husk, wheat straw, peanut powder, pomace, corn cobs, rice straw, sawdust and sugarcane bagasse for the cost-effective production of xylanase by *Bacillus safensis* XPS7 using the one-variable-at-a-time approach. A number of bacterial strains were isolated from different locations in the cold desert region of Himachal Pradesh, India. Among these, the hyperproducing strain designated as XPS7 was selected for optimized production of xylanase and identified as *B. safensis* based on 16S rDNA gene analysis. *B. safensis* XPS7 produced the maximum xylanase (141.28 U/mL) at 45 °C, pH 9, 24 h incubation time, 2% (*w/v*) wheat straw and wheat bran mixture as carbon source and 1.5% (*w/v*) ammonium nitrate as the nitrogen source in modified Riviere's medium. The results showed that the combination of wheat straw and wheat bran proved to be a cheap and abundant source for the hyper production of xylanase and can be used as an inexpensive base (carbon source) for large scale industrial production of enzymes. In addition, the use of waste for the economic production of enzymes will also help to minimize the environmental problems associated with the disposal of agro-industrial waste.

**Keywords:** agro-industrial wastes; carbon source; inexpensive; optimization; xylanase

## 1. Introduction

Agricultural and industrial wastes are the most abundant renewable resources in the biosphere and are available in large quantities. However, the disposal of these wastes causes serious health and environmental problems. Recently, potential efforts have been directed toward the utilization of these wastes as raw materials for the production of value-added products such as microbial enzymes through solid-state or submerged fermentation systems [1,2]. In previous studies, various forestry and agro-industrial wastes such as sugarcane bagasse, wheat bran, rice bran, rice straw, wheat straw, paper industry wastes, and corn cobs have been used for the production of xylanolytic enzymes and other value-added products [3,4].

Xylanases are one of the major hydrolytic enzymes involved in the degradation of hemicellulases, especially xylan. Xylanases are produced by various organisms such as microorganisms [5–19], marine algae, protozoa, snails, crustaceans, insects and seeds, etc. [4,20,21]. Xylanases are mainly produced by microorganisms, e.g., bacteria [5–12], filamentous fungi [5,13–15], actinomycetes [16,17], and yeasts [18,19]; although fungal xylanases initially attracted much interest in various industrial fields, bacterial xylanases were later preferred because they grow faster, require less space, are easy to maintain, and can be genetically manipulated [16].

Xylanolytic enzymes from microbes have attracted considerable attention in the current biotechnological era with their extensive applications in several industrial processes such as food, feed, pulp and paper, leather, detergent, pharmaceutical, textile and biofuel

industries [4,22,23] but their application is limited, owing to the high cost of fermentation medium and processes which accounts for 30–40% of the production cost. Therefore, it is extremely important to explore cheap alternative substrates for the production of xylanase.

The use of agro-industrial residues as a carbon source for industrial xylanase production has been limited and not extensively studied [24]. Although most studies on xylanase production have been conducted in submerged liquid cultures, there are a few reports on xylanase production in solid state fermentation using agricultural residues [1–3,24]. Submerged fermentation processes are generally preferred because they offer better nutrient availability, adequate oxygen supply and less time for fermentation compared to other fermentation systems.

The appropriate selection of an ideal agro-biotech waste or waste mixture for enzyme production in a submerged fermentation process depends on a plethora of physico-chemical parameters, generally related to the cost and availability of the substrate material, and may therefore involve screening of different agro-industrial residues [1–3,22,25]. Enzyme production using agro-industrial wastes as substrates in various fermentation systems has brought a whole range of benefits in terms of productivity, cost efficiency, time and media components. In addition, it can also help to reduce the pollution load from the environment.

Considering these facts, the present investigation aimed to utilize low-value substrates such as agro-industrial wastes for the production of xylanase by the bacterial isolate XPS7 in the context of submerged fermentation, which will ultimately reduce the production cost of the enzyme on a commercial scale and provide an economically viable alternative for the utilization of waste biomass.

## 2. Materials and Methods

### 2.1. Microorganism

The most active xylanase producing bacterial isolate XPS7 used in the present study was isolated from the cold desert area of H.P. in Research Lab-V, Department of Biotechnology, Himachal Pradesh University, Shimla. The culture was revived, maintained on nutrient agar slants and stored at 4 °C for further use.

### 2.2. Molecular Identification of the Potent Xylanase Producer

Identification of the potential isolate XPS7 at the gene level was performed using 16s rRNAgene analysis. The genomic DNA of the bacterial isolate was isolated using a DNA extraction solution and then qualitatively analyzed on 1.0% agarose. Amplification of the ~1.5 kb 16S rDNA fragment was performed using Taq DNA polymerase and the forward (27F) and reverse (1492R) primers. Isolate identification was confirmed by sequencing and phylogenetic analysis. The 16s rRNA sequence analysis confirmed that XPS7 had the maximum similarity to the *Bacillus safensis* strain (Supplementary Table S1 and Figure S1).

### 2.3. Production of Xylanase by Bacillus safensis XPS7

A loopful of bacterial culture was inoculated in 50 mL of nutrient broth and incubated at 30 °C under shaking conditions (120 rpm) for 24 h. An amount of 4% of the 12 h old inoculum was inoculated into 50 mL of production medium (modified Riviere's medium) containing: Xylan (0.5 g), $KH_2PO_4$ (0.69 g), $(NH_4)_2SO_4$ (0.69 g), $NaNO_3$ (0.1 g); $CaCl_2$ (0.01 g), $MgSO_4$ (0.02 g), yeast extract (0.05 g), trace element solution containing 0.15% $FeSO_4 \cdot 7H_2O$, 0.12% $MnCl_2 \cdot 4H_2O$ and incubated at 30 °C for 24 h at 150 rpm under shaking conditions. The culture contents were centrifuged at 11,739× *g* for 10 min at 4 °C and resulting supernatant was further assayed for xylanase activity under standard assay conditions.

Xylanase Determination (Analytical Method)

Xylanase activity in the supernatant was determined using the spectrophotometric method described by Bailey et al. [3] and Miller [26]. For this purpose, 0.5 mL of xylan solution and 0.3 mL of citrate buffer (0.55 M, pH 5.0) were added followed by 0.2 mL of

enzyme solution and incubated at 45 °C for 15 min. After cooling to room temperature, 3 mL of DNS reagent was added to each test tube to stop the reaction and absorbance (OD) was measured at 540 nm against a blank. The concentration of reducing sugar was calculated using a standard curve of D-xylose (20–200 µg xylose/mL). One unit of xylanase enzyme corresponds to the amount of enzyme releasing 1 µ mole of reducing sugar (xylose) per min per mL of culture supernatant under standard assay conditions.

### 2.4. Screening of Different Agro-Industrial Waste Substrates for Xylanase Production

Various agro-residues (wheat bran, wheat husk, wheat straw, peanut powder, pomace, corn cob, rice straw, sawdust and sugarcane bagasse, and in different combinations) were screened for extracellular xylanase production by *B. safensis* XPS7. Commercially available substrate, i.e., birchwood xylan was used as a control for comparative studies. All agro-industrial wastes were dried and finely ground in an electric grinder and added to the selected medium individually and in various combinations. Enzyme activity was measured in each case and the selected substrate was used for further experiments.

### 2.5. Optimization of Production Parameters

All fermentation experiments to optimize production conditions such as different growth media, different nitrogen sources, pH, temperature, age and size of inoculum, incubation time, etc. were performed under shake-flask conditions in 250 mL Erlenmeyer flasks containing 50 mL of production medium and then an enzyme assay was performed in each case to ascertain the effects of these culture conditions on enzyme yield.

### 2.5.1. Selection of Media

Selected agro-industrial residues were used to grow the bacterial strain on the following eleven different media to select the best growth medium for enzyme production (see Supplementary Table S2). The best selected medium (M5) was used for further studies.

### 2.5.2. Carbon Source Concentration

To select the preferred concentration of carbon source, 50 mL of the production medium with different concentration of substrate (i.e., 0.5% to 5%) was inoculated with the seed culture and incubated at 30 °C for 24 h to study the effects on enzyme activity. The supernatant collected after 24 h of incubation was used to determine xylanase activity.

### 2.5.3. Nitrogen Source

Various organic and inorganic nitrogen sources were used to check their effects on xylanase enzyme production. Tryptone, malt extract, urea, ammonium nitrate, ammonium sulphate, meat extract, beef extract and yeast extract along with peptone (1%) were added to 50 mL of production medium along with peptone (1%) and incubated at 30 °C for 24 h. After incubation, the supernatant was used for xylanase assay.

### 2.5.4. Nitrogen Source Concentration

To observe the optimization of nitrogen concentration, 50 mL of production medium containing different concentrations of ammonium nitrate (0.5% to 4%) was inoculated with the seed culture and incubated at 30 °C for 24 h. After incubation, cells were separated by centrifugation and xylanase activity was determined.

### 2.5.5. pH

The pH of the 50 mL production medium was adjusted to different values between 5.0 and 9.5 with 0.1 N NaOH or 0.1 N HCl before sterilization and inoculated with the seed culture and incubated at 30 °C for 24 h to determine the effect of pH on bacterial growth. Flasks containing 50 mL of the medium were adjusted to various pH values and xylanase activity was determined after incubation.

### 2.5.6. Incubation Temperature

For this purpose, 250 mL Erlenmeyer flasks, each containing 50 mL of the production medium were inoculated with the seed culture. The flasks were incubated at different temperatures ranging from 20, 25, 30, 35, 40, 45, 50, 55 and 60 °C for 24 h. After incubation for 24 h, xylanase activity was determined.

### 2.5.7. Inoculum Age

For this purpose, the loop full of bacterial culture was inoculated in 50 mL of seed medium and incubated at 30 °C in a shaking incubator. After 2, 4, 8, 10, 12, 16, 18, 20, 22 and 24 h, 1 mL of sample was inoculated from the seed medium into the production medium and incubated at 30 °C for 24 h. The culture broth was harvested, and the enzyme assay was performed. The inoculum that showed the highest xylanase activity was used for further studies.

### 2.5.8. Inoculum Size

Different inoculum sizes ranging 1–10% were used to study the effects of inoculum volume on enzyme activity. The 50 mL Erlenmeyer flasks containing the production medium were incubated at 30 °C for 24 h in a rotary shaker at 150 rpm and the cells were separated by centrifugation and xylanase activity was determined.

### 2.5.9. Production Time

The different parameters previously optimized were kept constant and 50 mL of production medium was inoculated with 2% seed culture. The Erlenmeyer flask was then incubated at 30 °C and the sample was taken after an interval of 4 to 52 h. Simultaneously, the culture was harvested, and xylanase activity was assayed.

### *2.6. Optimization of Reaction Parameters*

The assay conditions play a very important role in the enzyme activity. To obtain the maximum xylanase activity of XPS7, various reaction parameters were optimized by considering one variable at a time approach.

### 2.6.1. Buffer and Buffer pH Value

To find the appropriate buffer for the xylanase assay, the enzyme activity was assayed with different buffers and their pH values, i.e., Tris HCl (pH 7–9), sodium phosphate buffer (5–8), citrate buffer (3–6), carbonate buffer (9–10.4) and glycine-NaOH (7–10). An amount of 1 mL of the reaction mixture with the appropriate buffer was incubated for 15 min and the sugar estimation was performed by DNSA method.

### 2.6.2. Buffer Molarity

The molarity of the citrate buffer was varied in a range between 0.05 M, 0.1 M, 0.15 M, 0.2 M, 0.25 M, 0.3 M, 0.35 M, 0.4 M, 0.45 M and 0.5 M to optimize the best working molarity of pH 5.4. After the assay, the best buffer molarity was used for the next step.

### 2.6.3. Substrate Concentration

The substrate concentration was varied between 0.5% and 3% to determine the best substrate concentration to effectively perform the enzyme assay. All previously optimized parameters were kept constant. The reaction cocktail was incubated for 15 min and xylanase activity was estimated.

### 2.6.4. Reaction Time

The reaction time for the reaction was varied from 0 to 21 min by stopping the reaction every 2 min and the optical density was measured at 540 nm after performing the xylanase assay.

### 2.6.5. Reaction Temperature

For the enzyme activity assay, the reaction mixture was incubated at different temperatures, i.e., 10 °C, 20 °C, 30 °C, 40 °C, 50 °C, 60 °C, 70 °C, 80 °C and room temperature. After an incubation period of 8 min, optical density was measured to determine the sugar content using DNSA reagent.

### 2.7. Statistical Analysis

All experiments were performed in triplicate and the values presented were the mean values of triplicate. The data obtained after the experiments were statistically evaluated using ANOVA with a Tukey post hoc test using GraphPad Prism (version 8.0) Software (San Diego, CA, USA). The differences between the means were considered significant when the calculated $p$-values were $p < 0.05$ ** and $p < 0.01$ *.

## 3. Results and Discussion

The economic efficiency of any process is the key factor, especially for industrial production. The cost of raw material and downstream processing are also crucial factors for the success of a product on a larger scale. Therefore, the selection of an appropriate agro-residue (as a carbon source) is a crucial step for the economic production of enzymes, as it affects the final cost of the product. The use of abundantly available agro-residues as carbon sources in fermentation processes serves a dual role, i.e., cost-effective enzyme production and environmental safety. In order to cut down the production cost of the enzyme, this study investigated various abundantly available agro-residues as substrates (individually and in different combinations) for xylanase production.

### 3.1. Screening of Different Agro-Residues for Xylanase Production

In the present study, various low-cost and abundantly available agro-residues were screened individually and in different combinations for their suitability for xylanase production (Figure 1A,B). Among the various agro-residues used, wheat straw was found to be the best carbon source for xylanase production, followed by wheat bran and wheat husk (Figure 1A). Ho [1] also screened different agro-wastes and found that wheat bran was a good substrate for xylanase production by *Bacillus subtilis*. In addition, different combinations of agro-wastes were used as substrates for xylanase production, which indicated that a combination of wheat straw and wheat bran in a 1:1 ratio emerged as the best substrate for xylanase production (29.11 U/mLs) compared to individual agro-wastes and also instead of pure xylan (28.34 U/mL) (Figure 2). Ghoshal and co-workers [11], who used various lignocellulosic biomass for xylanase production, reported that wheat bran and wheat straw were the best substrates for xylanase production. In another study, Taie et al. [27] showed that *Trichoderma hamatum* showed a remarkable increase in xylanase production when grown on a medium containing wheat straw and rice husk as the sole carbon sources. Hence, a combination of these two low cost agro-residues were selected as substrate for optimized xylanase production.

### 3.2. Optimization of the Nutrient Medium Containing Agricultural Waste for Xylanase Production

The growth medium plays a very important role in the optimal growth of microorganisms and the subsequent release of enzymes and resulting products. In the present study, out of the total eleven different media tested, the highest xylanase activity of *B. safensis* XPS7 was detected in medium M5, i.e., modified Riviere medium with agro-residues as carbon source (32.54 U/mL) and the lowest activity was recorded in M1 (2.03 U/mL) (Figure 3). Previous studies reported that Riviere medium is the best medium for xylanase production with *Alternaria* sp. [28]. Since medium M5 was found to be the best, this growth medium was selected for future experiments.

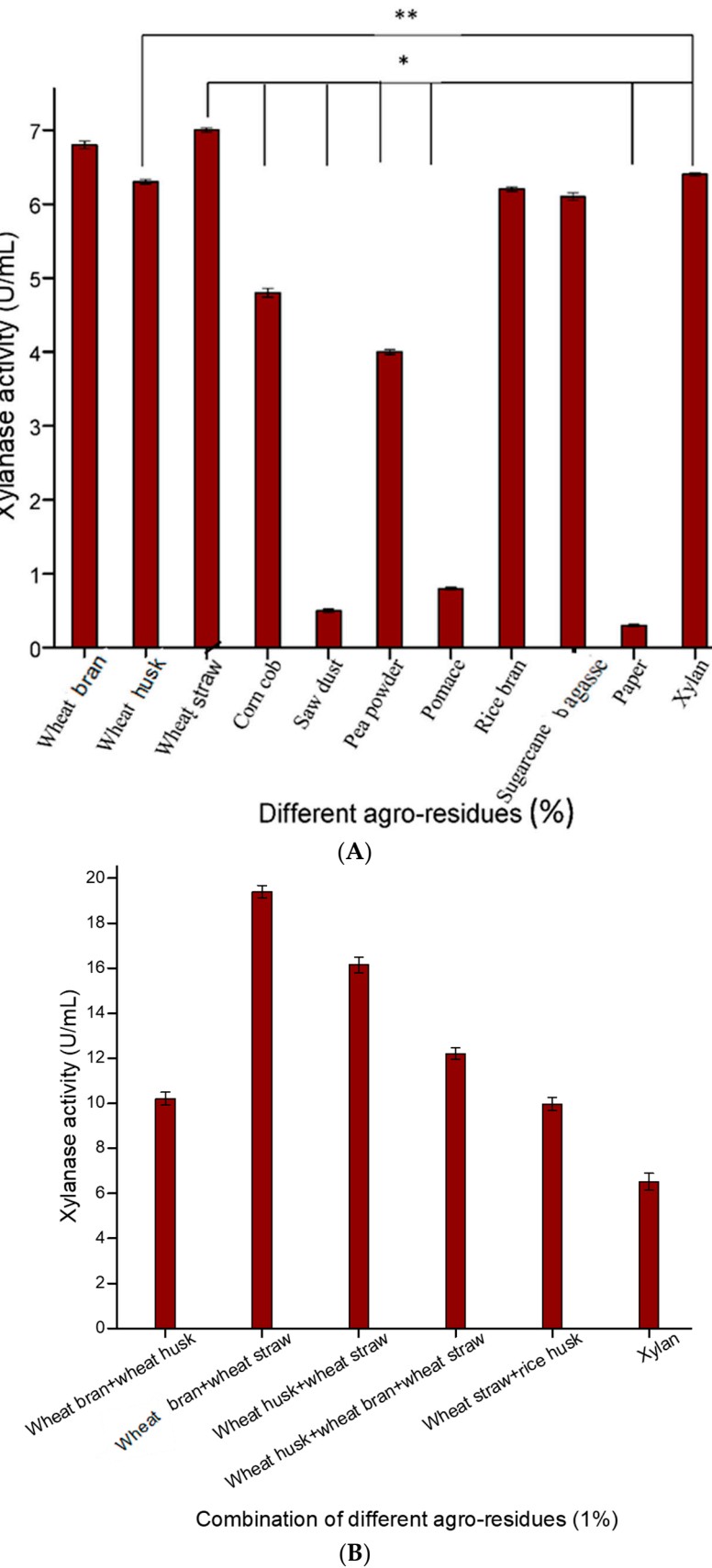

(A)

(B)

**Figure 1.** (**A**) Screening of different types of agro-residues for the production of xylanase by *B. safensis* XPS7. All the experiments were performed in triplicate. The results are expressed as mean ± SD and statistical analysis was performed using a one-way ANOVA with a Tukey post hoc test. Differences

between means were considered to be significant if the calculated *p*-values were $p < 0.05$ ** and $p < 0.01$ *. (**B**) Screening of different types of agro-residues in combination for the production of xylanase by *B. safensis* XPS7. All the experiments were performed in triplicate. Bars display mean $\pm$ SD and statistical analysis was performed using a one-way ANOVA with a Tukey post hoc test, with $p < 0.05$ ** and $p < 0.01$ *.

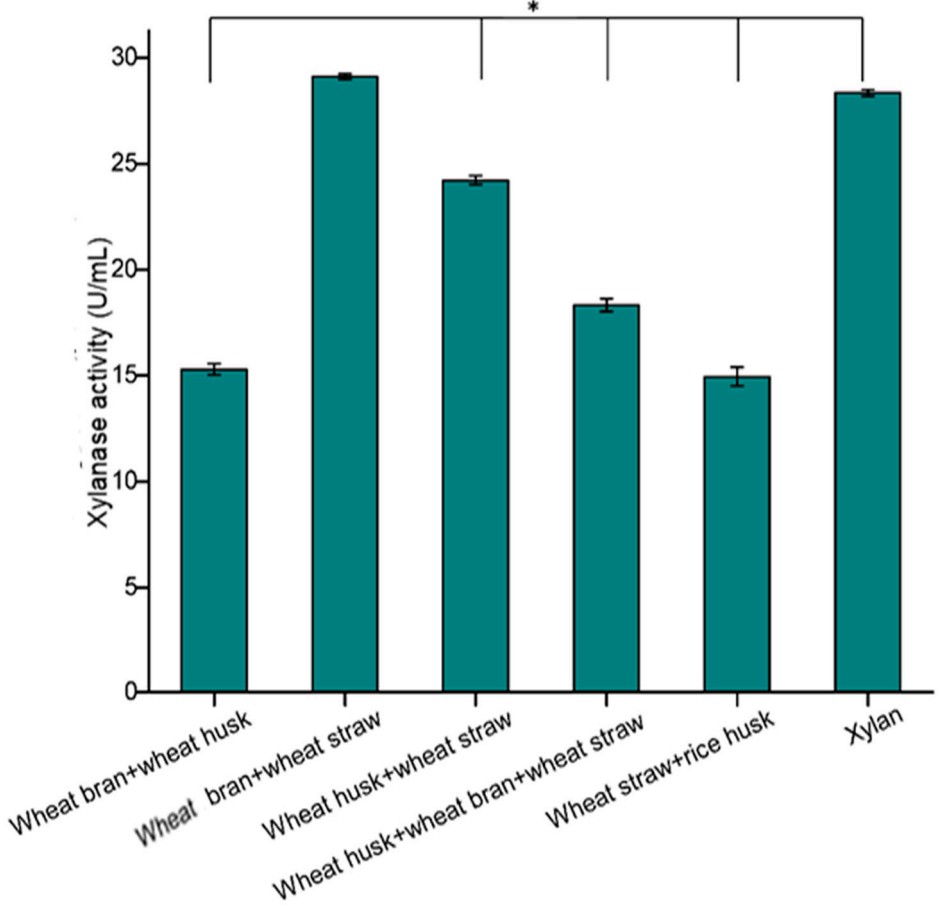

**Figure 2.** Screening of different types of agro-residues in combination for the production of xylanase by *B. safensis* XPS7. All the experiments were performed in triplicate. Bars display mean $\pm$ SD and statistical analysis was performed using a one-way ANOVA with a Tukey post hoc test, with $p < 0.01$ *.

### *3.3. Optimization of Process Parameters for Xylanase Production*
### 3.3.1. Optimization of Carbon Source Concentration

It has been reported that the type and concentration of carbon sources greatly affect enzyme production. In the present study, varied concentrations of wheat bran and wheat straw were tested and maximum production (38.89 U/mL) was obtained with 2% substrate (Figure 4). However, when the concentration of wheat bran and wheat straw was increased beyond 2%, a gradual decrease in enzyme activity was observed (Figure 4). This decrease in xylanase production could be due to the increase in the viscosity of the fermentation medium, which ultimately has a negative effect on the uniform circulation of nutrients and oxygen, thus decreasing microbial growth, leading to a decline in endo-I, 4-β-xylanase production [29,30]. In a previous study, Li et al. [31] reported a substrate (wheat bran) concentration of 4.93% for maximum production of xylanase, whereas much less substrate was used in the present study.

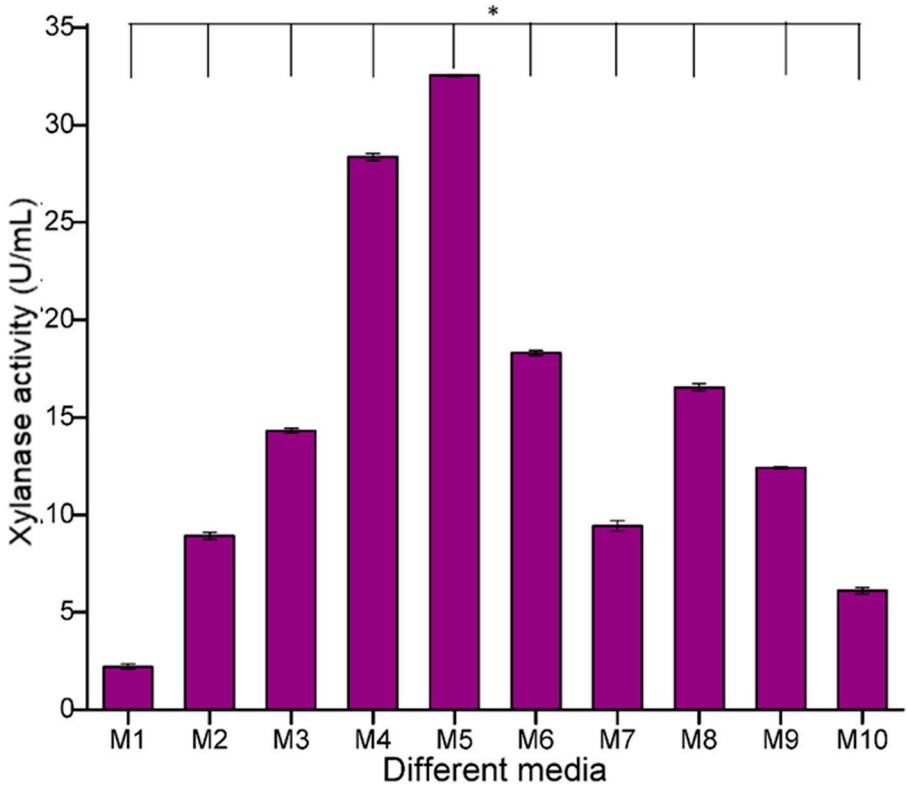

**Figure 3.** Screening of different media for xylanase production by *B. safensis* XPS7. All experiments were performed in triplicate. Bars display mean ± SD and statistical analysis was performed using one-way ANOVA with Tukey post hoc test, with $p < 0.01$ *.

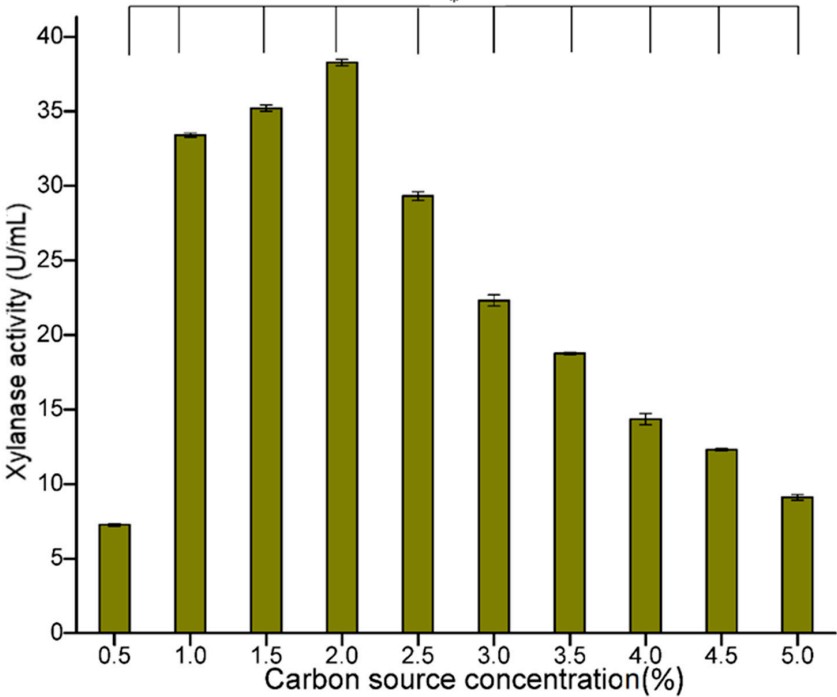

**Figure 4.** Effect of carbon source concentration on xylanase production by *B. safensis* XPS7 (Medium: M5; temp.: 30 °C; time: 24 h; pH: 8; inoculum: 4%). All the experiments were performed in triplicate. Bars display mean ± SD and statistical analysis was performed using a one-way ANOVA with a Tukey post hoc test, with $p < 0.01$ *.

### 3.3.2. Optimization of Different Nitrogen Sources

Another important element for enzyme production is nitrogen, which is present in many organic compounds, including amino acids and as part of nucleotide bases. Nitrogen makes up about 14% of the dry weight of microbial cells. Among the nitrogen sources used in the present study, inorganic nitrogen sources showed increased production compared to organic nitrogen sources. In this study, ammonium nitrate proved to be the best nitrogen source with maximum xylanolytic activity (74.68 U/mL), followed by ammonium sulphate (64.21 U/mL) (Figure 5). Our results are in agreement with those of [32], who reported increased production of xylanase by *Aspergillus tamari* when ammonium nitrate was used as an inorganic nitrogen source. Mehta et al. [33] also reported that ammonium nitrate was the best inorganic source for increased xylanase production by *Aspergillus* sp. It was also observed that most of the organic nitrogen sources tested resulted in a decrease in the enzyme activity compared to the control, with the lowest activity recorded in peptone (7.38 U/mL). However, when ammonium nitrate was used as a nitrogen source, the enzyme activity increased significantly (68.24 U/mL), followed by ammonium sulphate (64.21 U/mL) (Figure 5).

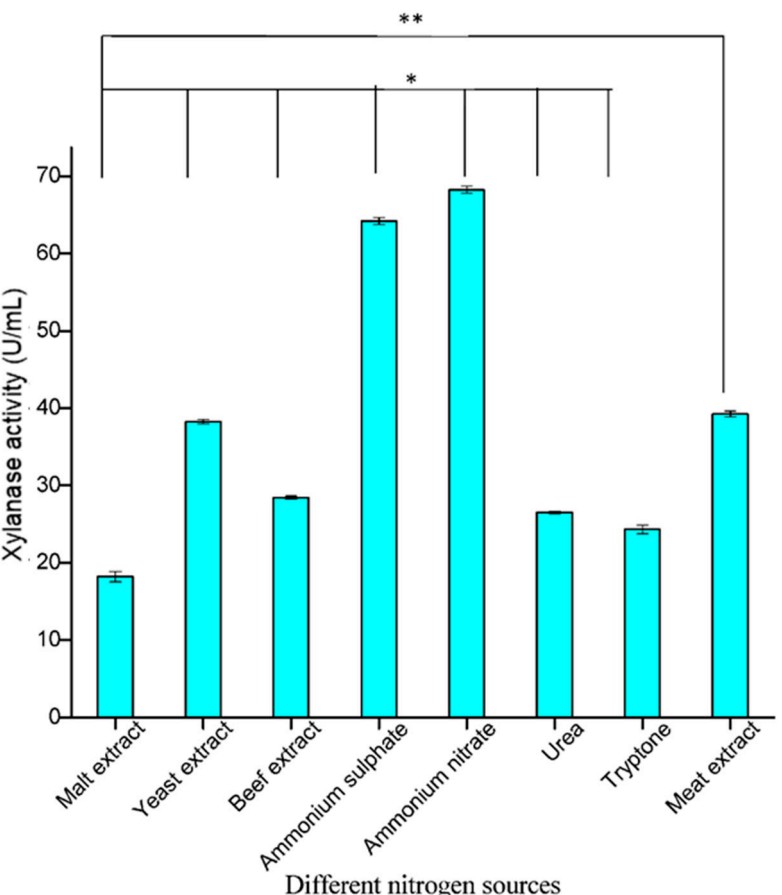

**Figure 5.** Effect of nitrogen sources on the production of xylanase by *B. safensis* XPS7 (Medium: M5; temp.: 30 °C; time: 24 h; pH: 8; inoculum: 4%). All the experiments were performed in triplicate. Bars display mean ± SD and statistical analysis was performed using a one-way ANOVA with a Tukey post hoc test, with $p < 0.05$ ** and $p < 0.01$ *.

### 3.4. Optimization of Varied Concentrations of Ammonium Nitrate

*B. safensis* XPS7 was cultured on a production medium with varying concentrations of ammonium nitrate (i.e., 0.5–4% *w/v*) to check the effects on relative enzyme yield. The results of xylanase activity as shown in Figure 6 revealed that the maximum xylanase activity of 71.84 U/mL was attained at 1.5% ammonium nitrate, but thereafter decreased with increasing ammonium nitrate concentration. A uniform decline in xylanase activ-

ity was observed when the concentration of ammonium nitrate was increased from 2% (68.31 U/mL) to 4% (10.22 U/mL).

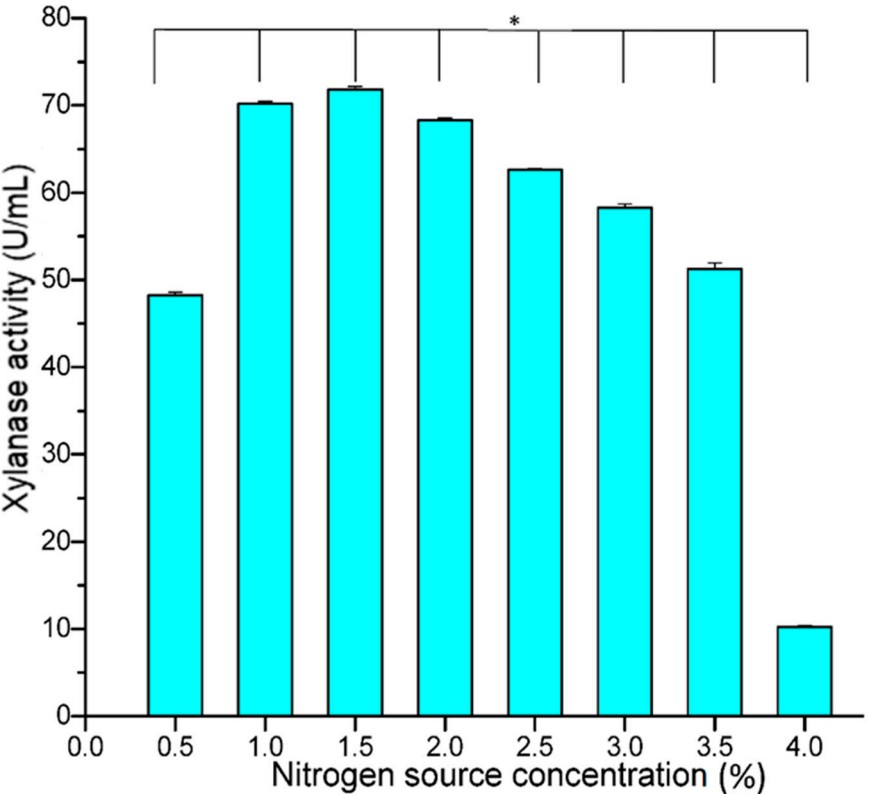

**Figure 6.** Effect of nitrogen source concentration on the xylanase production by *B. safensis* XPS7 (Medium: M5; temp.: 30 °C; time: 24 h; pH: 8; inoculum: 4%). All the experiments were performed in triplicate. Bars display mean ± SD and statistical analysis was performed using a one-way ANOVA with a Tukey post hoc test, with $p < 0.01$ *.

### 3.5. Optimization of Physical Parameters

3.5.1. Effect of pH

The influence of pH on xylanase activity of crude enzyme showed that maximum xylanase production was observed at pH 9.0 (73.42 U/mL) and least at pH 12 (10.39 U/mL). Earlier studies have also revealed similar results where the maximum xylanase activity was reported at the same pH for *Arthrobacter* sp. MTCC 6915 [34]. However, enzyme production decreased with an increase in pH after 9 (Figure 7). This decrease in xylanase production could be due to changes in the 3-dimensional structure of the proteins at different pH levels, leading to denaturation of the enzymes [35] (Uday et al., 2016).

3.5.2. Effect of Temperature

The effect of temperature on the release of extracellular xylanase by the bacterial isolate was studied and maximum xylanase was measured at 45 °C (74.89 U/mL), suggesting that the organism may be thermophilic. Xylanase activity remained stable over a wide temperature range, i.e., in the range 35–55 °C (Figure 8). The temperature of 50 °C had little influence on the enzyme activity, but further increases beyond 50 °C showed a decline in enzyme activity, with the least activity of 36.25 U/mL at 60 °C. This character of the organism can be better utilized on an industrial scale. In earlier studies, Amutha et al. [36] reported an optimum temperature of 50 °C for xylanase production. At 50 °C, the activity of the enzyme decreased due to the denaturation of the enzyme protein [37]. Irfana and co-workers [2] also reported an optimal temperature of 50 °C for xylanase production by *B. subtilis* BS04 and *B. megaterium* BM07, respectively.

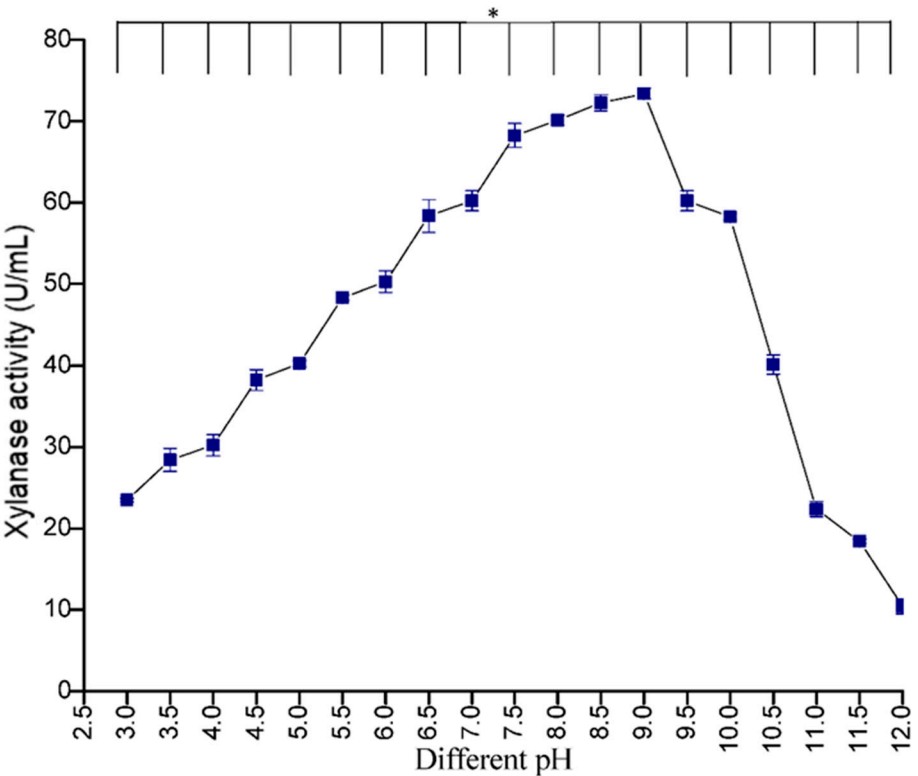

**Figure 7.** Effect of pH on xylanase production by *B. safensis* XPS7 (Medium: M5; temp.: 30 °C; time: 24 h; pH: 2.5 to 12; inoculum: 4%). All the experiments were performed in triplicate. Line graph display mean ± SD and statistical analysis was performed using a one-way ANOVA with a Tukey post hoc test, with $p < 0.01$ *.

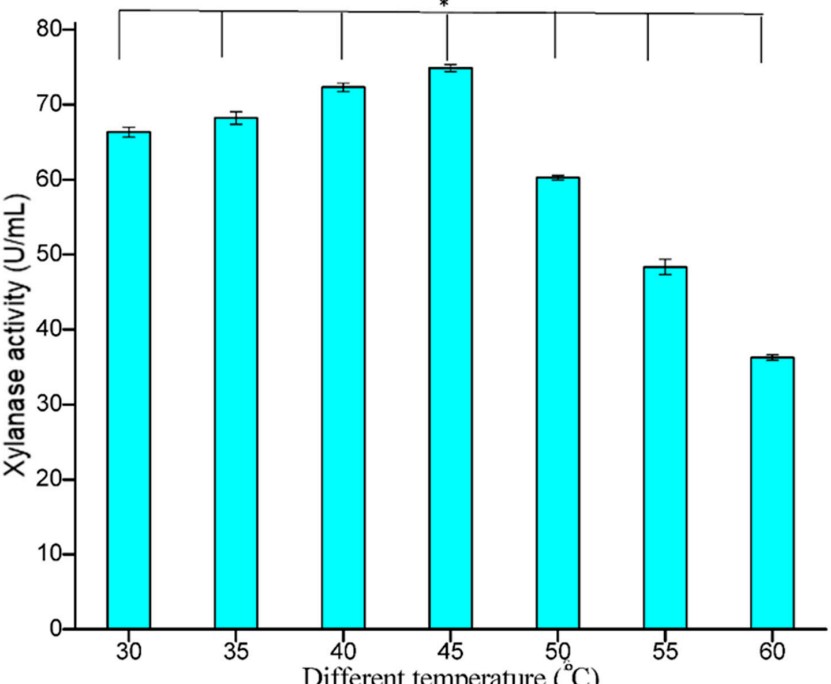

**Figure 8.** Effect of temperature on the production of xylanase by *B. safensis* XPS7 (Medium: M5, temp.: 30 to 60 °C; time: 24 h; pH: 9; inoculum: 4%). All the experiments were performed in triplicate. Bars display mean ± SD and statistical analysis was performed using a one-way ANOVA with a Tukey post hoc test, with $p < 0.01$ *.

### 3.5.3. Effect of Inoculum Age and Size

The age of the inoculum plays an important role in determining the viability of the cells and also in metabolism, which affects the production of enzymes. In the present study, maximum enzyme production (73.49 U/mL) was recorded in 8 h-old culture, and thereafter a decrease in enzyme activity was observed (Figure 9). The size of the inoculum also influenced the production of enzymes. In the present study, 4% inoculum size was found to be optimum for xylanase production (Figure 10). Irfana et al. (2016) also reported maximum xylanase production at 4% inoculum.

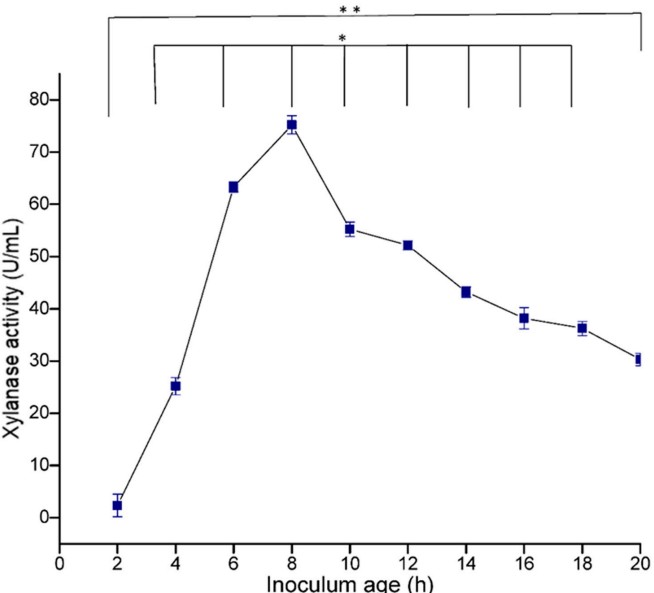

**Figure 9.** Optimization of inoculum age for the production of xylanase by *B. safensis* XPS7. (Medium: M5; temp.: 45 °C; time: 24 h; pH: 9; inoculum: 4%). All the experiments were performed in triplicate. Line graph display mean ± SD and statistical analysis was performed using a one-way ANOVA with a Tukey post hoc test, with $p < 0.05$ ** and $p < 0.01$ *.

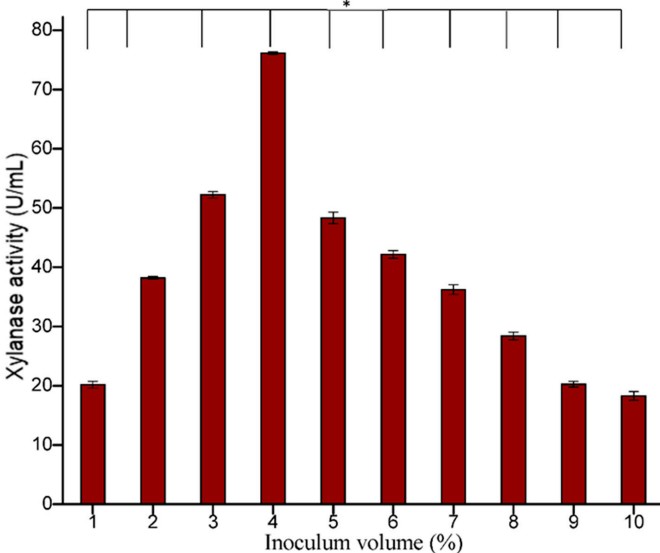

**Figure 10.** Optimization of inoculum size for the production of xylanase by *B. safensis* XPS7 (Medium: M5; temp.: 45 °C; time: 24 h; pH: 9; inoculum: 1 to 10%). All the experiments were performed in triplicate. Bars display mean ± SD and statistical analysis was performed using a one-way ANOVA with a Tukey post hoc test, with $p < 0.01$ *.

### 3.5.4. Effect of Incubation Time

The production of an enzyme also greatly depends upon incubation time. In the present study, maximum xylanase activity (75.86 U/mL) was recorded after 24 h of incubation (Figure 11). The findings of the present work are in agreement with several earlier reports, including Heck et al. [38] and Simphiwe et al. [39], who also found that 24 h incubation was best for maximum production of xylanase.

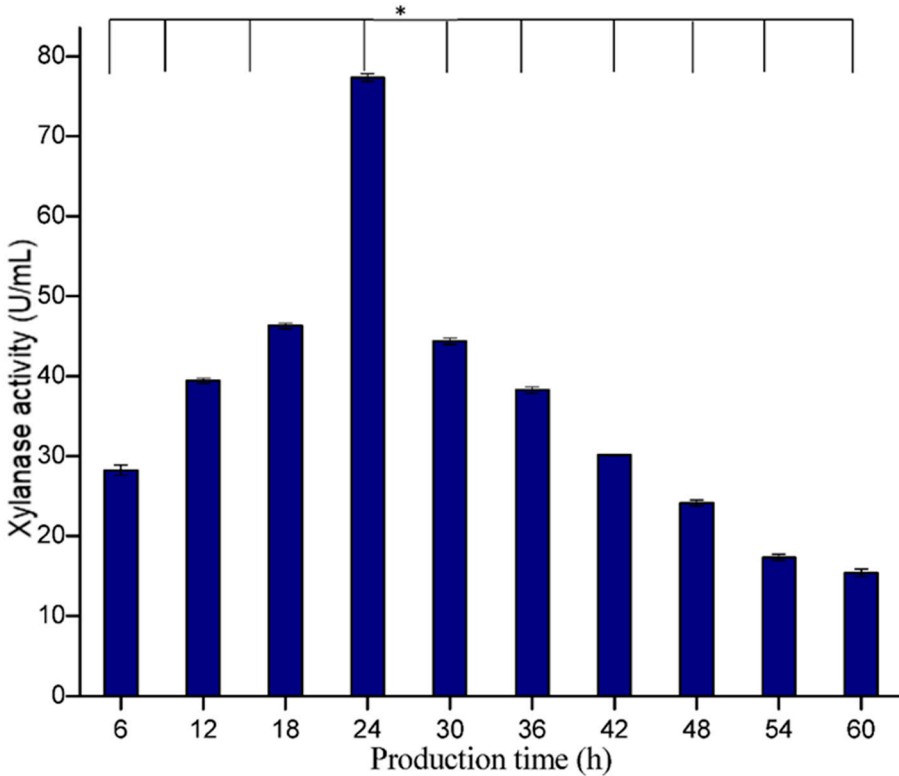

**Figure 11.** Time course of xylanase production by *B. safensis* XPS7 (Medium: M5; temp.: 45 °C; time: 6 to 60 h; pH: 9; inoculum: 4%). All the experiments were performed in triplicate. Bars display mean ± SD and statistical analysis was performed using a one-way ANOVA with a Tukey post hoc test, with $p < 0.01$ *.

### 3.6. Optimization of Reaction Conditions

In addition to the production conditions, the reaction conditions were also optimized. The buffer and its pH are critical for enzyme reaction and activity. Enzyme activity is markedly affected by pH because substrate binding and enzyme catalysis are often dependent on the charge distribution of both the substrate and the enzyme molecule [40]. In the present study, sodium citrate buffer (50 mM) with a pH of 6.0 was found to be most suitable for xylanase activity of the bacterial isolate, indicating that the enzyme works best under neutral conditions (Supplementary Figures S2 and S3). Amutha et al. [36] reported maximum xylanase activity at pH 6.5, which was very near to the value obtained in the present study. It was also observed that the enzyme activity decreased with increasing substrate concentration and the maximum enzyme activity was obtained with xylan concentration of 1% (Supplementary Figure S4). Higher or lower substrate concentrations resulted in a decrease in enzyme activity, which could be due to a decrease in the binding efficiency of the enzyme. In the present study, the maximum activity (81.32 U/mL) was observed at a substrate concentration of 1% substrate, which is consistent with the findings of Bhakyaraj [41].

The influence of reaction time was also studied with the bacterial xylanase, and the maximum activity was recorded at 9 min (140.17 U/mL) (Figure 12). With further increase in the incubation time, the enzyme activity decreased, which might be due to the denaturation

of the enzyme as a result of the prolonged incubation. The findings of the present work are in agreement with the reports of Faulet et al. [42], Sharma and Sharma [43] and some others, who considered a reaction time of 10 min as optimal for bacterial xylanase activity. Each enzyme has a specific temperature range in which it can work actively. Outside this range, the enzyme becomes inactive. This is because temperature changes led to result in the breakdown of intermolecular bonds (H-bond, dipole–dipole interaction) between polar and hydrophobic forces between nonpolar groups in the protein. The influence of temperature on xylanase activity was studied, and maximum activity was achieved at 55 °C (141.23 U/mL) (Supplementary Figure S5). Kamble and Jadhav [44] reported maximum enzyme activity at 50 °C from *Aspergillus flavus*.

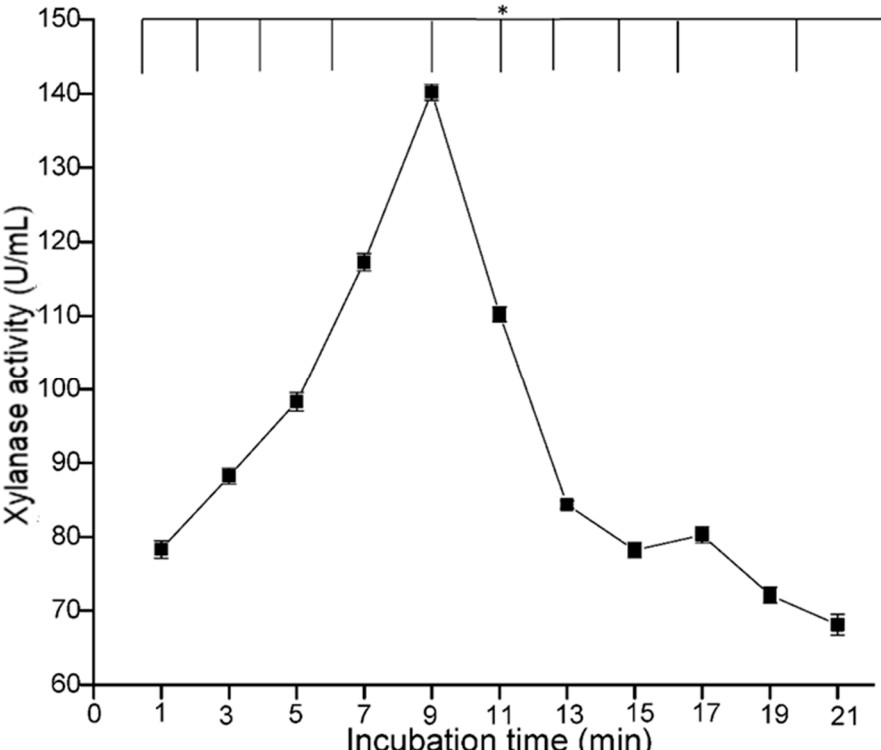

**Figure 12.** Effect of incubation time on xylanase production by *B. safensis* XPS7. All the experiments were performed in triplicate. Line graph display mean ± SD and statistical analysis was performed using a one-way ANOVA with a Tukey post hoc test, with $p < 0.01$ *.

The results indicated that a combination of wheat straw and wheat bran proved to be a cheap and abundantly available source for hyperproduction of xylanase. In this study, about a five-fold improvement in xylanase activity (141.28 U/mL) was achieved under the optimized conditions. Thus, enzyme production using agro-residues instead of expensive purified substrates opens new areas for large-scale and economic production of enzymes on an industrial scale. The ability of the organism and enzyme to withstand high pH, prolonged stability and require less inoculum speaks for the potential of the organism and enzyme. Further improvement of the strains and the use of purified enzymes can definitely improve the yield and reduce the overall production cost.

## 4. Conclusions

The present study aimed at the potential use of different cheap substrates for the production of xylanase by *B. safensis* XPS7. The results showed that a combination of wheat straw and wheat bran in a 1:1 ratio proved to be the best substrate for xylanase production and these two low-cost agro-residues were used as the sole carbon sources for optimal enzyme production. Besides the carbon source, other production conditions also play a significant role in enzyme activity. In this study, an approximately five-fold improvement in

xylanase activity (141.28 U/mL) was obtained under the optimized fermentation conditions with a temperature of 45 °C, pH of 9, incubation time of 24 h, a mixture of 2% (*w*/*v*) wheat straw and wheat bran as carbon source and 1.5% (*w*/*v*) ammonium nitrate as the nitrogen source in a modified Riviere's medium. The obtained results indicated that the combination of wheat straw and wheat bran was proved to be a cheap and abundantly available source for the hyperproduction of xylanase and can be used as an inexpensive base (carbon source) for large scale production of enzymes, which ultimately lowers the production cost of enzymes at one side, while on the other side, helps to minimize the environmental problems associated with the disposal of agro-industrial wastes.

**Supplementary Materials:** The following supporting information can be downloaded at: https://www.mdpi.com/article/10.3390/fermentation8050221/s1, Figure S1: Phylogenetic tree made using neighbour joining method. Figure. S2: Effect of different buffer systems and pH on the activity of xylanase by XPS7, Figure S3: Effect of buffer molarity on the production of xylanase by XPS7, Figure S4: Effect of substrate concentration on production of xylanase by XPS7, Figure. S5: Effect of incubation temperature on production of xylanase by XPS7, Table S1: Alignment view using combination of NCBI GenBank and RDP database, Table S2: Composition of different media used (g/L).

**Author Contributions:** S.D. did experimental work and wrote the manuscript. D.D. helped in experimental work of the study and analysis of results. A.K.B. supervised the research idea and helped in editing of the manuscript. The final version of the manuscript was read and approved by the authors. All authors have read and agreed to the published version of the manuscript.

**Funding:** The authors are grateful to the Ministry of Environment, Forest and Climate Change (MoEF&CC), Govt. of India, New Delhi, for financial support under 'National Mission on Himalayan Studies' (NMHS project no. GBPI/NMHS/HF/RA/2015-16) and Vice Chancellor, Himachal Pradesh University, Shimla for providing research facilities.

**Institutional Review Board Statement:** Not applicable.

**Informed Consent Statement:** Not applicable.

**Data Availability Statement:** Not applicable.

**Conflicts of Interest:** The authors declare no conflict of interest.

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
