# Peer review of "Utilization of Agroresidues for the Production of Xylanase by Bacillus safensis XPS7 and Optimization of Production Parameters"

_fermentation, doi:10.3390/fermentation8050221_

Round 1

Reviewer 1 Report

The authors describe the production of xylanase from a newly isolated Bacillus strain using different agro industrial wastes. The topic is interesting however it needs to be modified considering  the comments, and eventually other experiments need to be performed.

Major points:

There is a bit of confusion regarding the conditions in which experiments were performed and this needs to be clarified by the authors because it is difficult otherwise to draw conclusions. On which medium were experiments reported in Fig 1 and 2  conducted?

Why is xylanase activity so different on the control (xylan)  in figure 1 and figure 2? Were they tested in different conditions? Because looking at the values of enzymatic activity obtained on the different substrates compared to the control respectively in each experiment, it seems that a better improvement was obtained by using wheat straw or wheat bran  alone, not combined. If this is the case then I do not see the point of using both substrates in combination, and the authors should explain why they preferred using a combination of substrates that is giving worse results compared to single substrates.

After experiments reported in Fig 3 were all further trials performed on M5? This should be stated in the manuscript, together with the conditions in which all different experiments were performed (temp, pH, inoculum  etc)

Was the combination of all identified optimal conditions meaning pH, temperature, nitrogen source, inoculum age and inoculum size tested by the authors? For example the last experiment reported in Fig 11 was it performed by considering all the optimal parameters identified throughout the study? This should be clarified. If this is not the case the authors should add another set of experiments in which the optimal identified conditions are tested.

-How many replicates of each condition were performed? This should be written in the manuscript. At least three replicates for each experiments should be presented in order to identify significal differences.  A statistical analysis should be performed and significance of the results should be added for each set of experiments. The standard deviations are very low, considering the biological variability of this type of experiments.

The manuscript should be revised by a native speaker.

Minor points:

In the mat and methods section from Paragraph 2.3 onwards please specify if falcon tubes or shakeflasks or bottles were used for the experiments and their volume.

What is the percentage of waste used for experiments reported in Fig 1 and Fig 2?

Conditions in which experiments were performed (medium, temp, time, pH inoculm etc) should be reported in each figure legend, the figure needs to be self standing.

Reviewer 2 Report

The manuscript explained the production of xylanase by Bacillus safensis XPS7 using different agricultural wastes and several experiments have been carried out for screening the best substrate and optimization of different parameters for the production of xylanase. The work was good ,however, the manuscript needs some improvements and modifications as listed below:

Method part

  • Line 102: rpm should be changed to ×g and all ml should be written as mL.
  • The results may not be explained in the method part for example (Line 124: combination of wheat bran and wheat straw observed as best) or (Line 135: ammonium nitrate was found...)
  • Line 174 NaOH
  • Line 188 and 192: what is O:D?
  • Analytical methods were missing.

Results part

  • Line 207: From Fig.1, it was concluded that wheat straw was the best one not wheat bran.
  • Xylan was used as control in both figure 1 and 2. So the xylanase activity should be the same in both figures, but it is not. Why?
  • Fig.2 showed that combination of wheat bran and wheat straw was the best substrate, but then you used combination of wheat straw and rice husk which was a result of other study. Why didn't you use this combination in your experiments for Fig. 2 or why didn't you use your best combination result for further experiments?
  • Please write in the figure's caption how many replicates have you done for each experiment or figure.
  • Please specify the name of microorganism in Fig.1
  • Line 230: The lowest activity was M1 not M10 based on Figure 3.
  • Figure 4, 6: Axis x should have unit for the concentration.
  • Line 329: The best pH was 6 not 7 based on the figure SF2.

Round 2

Reviewer 1 Report

The authors addressed most of the comments. Minor points:

-By comparing Fig 1b and Fig 2 it seems that by increasing the concentration of substrate from 1 to 2% while on xylan there is a huge improvement of enzymatic activity, on wheat bran + straw  only a slight improvement of enzymatic activity is observed. The authors should comment on that.

-Be careful because in paragraph 2.3.1 you are using ml instead of mL.

-I still reccomend revision of the language from a native speaker, some sentecences need to be rewritten and I think it is in your interest to present a nicely written manuscript.
